# OpenReview forum: "Plan B: Training LLMs to fail less severely"
_ICLR.cc/2025/Conference — ICLR 2025 Conference Withdrawn Submission_

### Official Review · Reviewer_Uqay · 2024-11-03

**Soundness:** 2
**Presentation:** 2
**Contribution:** 2
**Rating:** 3
**Confidence:** 4

**Summary:**

The paper proposes an auxiliary safety mechanism for large language models (LLMs) called Plan B, which aims to reduce the severity of harmful responses in scenarios where failures cannot be prevented entirely. The idea is to focus on the fact that instead of solely relying on refusal for harmful requests, Plan B trains models to respond less harmfully when failures occur. The training involves preference-based learning to ensure refusals are prioritized and harmful responses are ranked by severity.

**Strengths:**

- The approach discusses an important point of prevention of harmful outputs to mitigate the severity of failures when they do occur. This perspective is practical given the current limitations in completely safeguarding models.

- The authors provide experimental results across various model sizes.

**Weaknesses:**

- While the idea seems interesting, I am not fully convinced that the proposed solutions make sense. Especially because the proposed solution seems to have a pseudo sense of safety than actual safety. The motivation part is not clear.

- Collecting the preference dataset for the training could add more challenges in the safety training.

- There are no theoretical insights about why the proposed approach is supposed to work or even any basic explanation of LLM safety.

- In other words, why the proposed solution makes sense is not clear.

- Is there an underlying alignment problem this paper is trying to solve?

**Questions:**

N/A

---

### Official Review · Reviewer_BcF6 · 2024-11-04

**Soundness:** 1
**Presentation:** 2
**Contribution:** 2
**Rating:** 3
**Confidence:** 4

**Summary:**

The primary idea is to design a mechanism to prevent harmful generations from the current LLMs. In order to do that, the paper rather than preventing harmful generations completely reduces their severity when they occur. Specifically, the LLM is trained to prefer refusals over less harmful responses over highly harmful ones and thereby showed the efficacy against backdoor attacks.

**Strengths:**

The paper proposes an interesting approach to not just directly optimize for the harmful response and optimize for refusal rather perform an intermediate step, by generating as a less harmless answer. The approach leverages odds ratio preference optimization to

**Weaknesses:**

1. The idea is intuitive, but the idea lacks any mathematical insights to what the justification of the improvement. It is not extremely clear, why allowing for less harmful responses will improve the performance of safety? It should depend on the evaluation metric. For example, if i take the standard harmless classifier/reward i.e. PKU/Anthropic benchmark, why this method should perform better?

2. How is the value lambda determined in equation 1? What's the convergence of this particular objective, in terms of where it will converge? How can we justify that this will improve the safety performance?

3. It will be helpful to rigorously describe or characterize the corresponding performance on other utilities based on this. Ideally, it should improve based on this idea right? Please have a discussion and clarification on that.

**Questions:**

See above.

---

### Official Review · Reviewer_Tu3F · 2024-11-04

**Soundness:** 2
**Presentation:** 2
**Contribution:** 2
**Rating:** 3
**Confidence:** 4

**Summary:**

This work explores the impact of adding “less harmful” responses in addition to the regular safety training pipeline, offering models an alternate safety measure that could complement existing approaches (i.e. models are trained to prefer harmless > less harmful > harmful). In the context of safety, “less harmful” responses could be vague information instead of detailed instructions for a given harmful request. The authors construct datasets with this property, and use Odds Ratio Preference Optimization to optimize for the ranked preference pairs, and compare it to a baseline method of only optimizing for refusals. LLM judges are used to evaluate the harm/helpfulness of responses. Reasonable reduction in harmfulness on backdoor attacks were shown (as scored by an LLM judge), and some transfer to jailbreak attacks is shown.

**Strengths:**

- Potentially promising results are obtained - reasonable improvements on their backdoor benchmarks (as scored by LLM judge models) given their problem setup, assuming 'less helpful' responses are considered to be acceptable
- The proposed method is compatible and simple to add to existing safety training pipelines
- Utility seems to be maintained across standard benchmarks

**Weaknesses:**

- While reasonable improvements have been shown for the backdoor attacks, there seems to be limited improvement to jailbreak robustness for many attacks (some reduction in often within error)
- Jailbreak evaluations are weak - PAIR was not directly used (only jailbreaks that were already generated against Llama-3 8B, no GCG)
- Overall technical contribution seems modest and fairly non rigorous, with no detailed analysis or explanations of why their method works is offered
- Average results in figure 1 could be misleading compared to the breakdown in figure 8

**Questions:**

- Why does training a model to prefer less harmful responses seem to work well on the backdoor attacks, but seems much less effective for jailbreak attacks? Is it possible that your first fine-tuning stage of injecting the backdoor is brittle and the second fine-tuning stage is simply undoing this first round of fine-tuning? There should be an ablation for this
- Are there other evaluations that could be done beyond using LLM judge models? The scores from these models can be subjective and opaque
- If I understand correctly, it seems like most of the harmful data focuses on requests which requires the LLM to respond with some informative/factual response, such as giving instructions on how to accomplish some task. Do you also evaluate whether models are less likely to produce toxic/hateful responses in general?
- In figure 1, why does refusal training make models *more* susceptible to jailbreaks?

---

### Official Review · Reviewer_FcJQ · 2024-11-06

**Soundness:** 3
**Presentation:** 3
**Contribution:** 3
**Rating:** 6
**Confidence:** 4

**Summary:**

This paper proposes "Plan B" - an approach to make language models fail more gracefully when their safety guardrails are bypassed. Rather than solely focusing on preventing harmful outputs, the authors train models to produce less harmful responses when safety measures fail. They evaluate this on backdoored models and show that Plan B training reduces the severity of harmful outputs while maintaining model capabilities. The key technique is training on preference orderings between refusal, less harmful responses, and more harmful responses. The authors demonstrate the approach generalizes to various jailbreak techniques beyond just backdoors.

**Strengths:**

Pros:
- The idea of having a second line of defense makes sense and in line with some recent previous works such as [Improving Alignment and Robustness with Circuit Breakers](https://arxiv.org/abs/2406.04313) and [Safety Alignment Should Be Made More Than Just a Few Tokens Deep](https://arxiv.org/abs/2406.05946).
- It’s nice to see novel ideas for alignment that go beyond simple refusals that are often easy to bypass via standard jailbreak techniques.
- The evaluation on MMLU and MT-Bench convincingly confirms that Plan B training doesn’t deteriorate standard performance.
- The paper is clearly written and proposes a novel idea for alignment.

**Weaknesses:**

Weaknesses:
- The experiments don’t seem very extensive. E.g., it would be good to include some additional ablation studies about the key hyperparameters of the proposed Plan B training.
- The choice of jailbreak methods seems a bit suboptimal. The authors seem to acknowledge that: *“We note that Crescendo and PAIR do not seem to produce particularly useful jailbreaks by default”*. One can try to include stronger jailbreaks, e.g., the template-based one from [Jailbreaking Leading Safety-Aligned LLMs with Simple Adaptive Attacks](https://arxiv.org/abs/2404.02151).

**Questions:**

Suggestions:
- [Safety Alignment Should Be Made More Than Just a Few Tokens Deep](https://arxiv.org/abs/2406.05946) has a similar high-level motivation and should be discussed.
- It would be valuable to discuss how this approach relates to the approach taken in the o1 models https://cdn.openai.com/o1-system-card-20240917.pdf. In particular, the safety training approach there suggests that the models should still comply with some harmful requests (or their reformulations) but shouldn’t disclose too critical/detailed information.
- Figure 3: the axes’ captions are too small.

---

### Official Review · Reviewer_vb8E · 2024-11-09

**Soundness:** 2
**Presentation:** 3
**Contribution:** 3
**Rating:** 5
**Confidence:** 4

**Summary:**

This paper suggests that we can train models to fail less severely using the simple idea that LLMs can learn the preference that less harmful (but compliant) response is better than a more harmful response. The implement this across a variety of settings and show that it improves robustness to jailbreaks and backdoors, without compromising on ability to solve benign tasks effectively.

**Strengths:**

I think this paper puts forward an interesting idea: jailbreaks might arise from the model having to choose between "complying" or "refusing". Several works have shown that simply preventing refusal is extremely challenging, and there are many jailbreaks. A hypothesis is put forward that maybe models can be taught to comply but provide less harmful information. The paper could make a better case for this idea in presentation, but overall it is a simple and clean idea. To the best of my knowledge, this brings a novel perspective to AI safety research. The paper shows empirical promise across a variety of LLMs.

**Weaknesses:**

While the idea is interesting, the paper lacks some rigor in evaluation and compelling explanation of some seemingly arbitrary choices made in evaluation. In particular, the role of the backdoor is unclear.

Careful evaluation is of utmost important in assessing safety - the paper uses existing jailbreaks and backdoors that haven't been "adapted" to the new defense. The paper doesn't make a compelling case that the evaluation is not subject to the issue of a false sense of security. A new attacker could train the backdoor to resist less-severe compliances and elicit severe harmful responses. Similarly, the optimization objective for jailbreaks could be modified to specifically find a very harmful compliance rather than a less harmful compliance. This might not be straightforward to do, but it merits some experiments and discussion.

The paper is also missing some key ablations in terms of sensitivity to how to construct preference data that effectively teaches a model to be "less" harmful. Is there a broadly reliable way to generate these "less" harmful responses - for it to be practical?

Finally, the LLM-as-judge evaluation for severity should be carefully audited to make sure that the evaluation can be trusted.

**Questions:**

(1)  Why are all the models considered (attacked via jailbreaks or backdoors) first back doored? Am I misunderstanding this? In particular, seems like backdoor should be an independent orthogonal evaluation, and we should evaluate success of jailbreaks against models that aren't backdoored as well.

(2) Can we consider new backdoors that know that plan B training is in place?

(3) Why does the plan B model occasionally comply to harmful responses that the baseline defense refuses? This seems like an interesting tradeoff to discuss.

(4) Similarly, for jailbreaks, is it possible to consider adaptive defenses that try to maximize an output which is harmful?

---

### Note · Authors · 2024-11-29

**Comment:**

We appreciate the reviewers’ feedback and acknowledge some valid concerns around evaluations on stronger (adaptive) jailbreaks as well as inclusion of more extensive ablations. While we believe that some reviewers underrated our existing contribution, we recognize the potential for an improved future submission.

**Withdrawal Confirmation:**

I have read and agree with the venue's withdrawal policy on behalf of myself and my co-authors.